# Increase in Phytoextraction Potential by Genome Editing and Transformation: A Review

**DOI:** 10.3390/plants11010086

**Published:** 2021-12-28

**Authors:** Javiera Venegas-Rioseco, Rosanna Ginocchio, Claudia Ortiz-Calderón

**Affiliations:** 1Departamento de Ecosistemas y Medio Ambiente, Facultad de Agronomía e Ingeniería Forestal, Pontificia Universidad Católica de Chile, Santiago 7820436, Chile; 2Center of Applied Ecology and Sustainability, Pontificia Universidad Católica de Chile, Santiago 8331150, Chile; 3Laboratorio de Bioquímica Vegetal y Fitorremediación, Departamento de Biología, Facultad de Química y Biología, Universidad de Santiago de Chile, Santiago 9160000, Chile; claudia.ortiz@usach.cl

**Keywords:** soil metal remediation, metallophytes, hyperaccumulators, phytoremediation

## Abstract

Soil metal contamination associated with productive activities is a global issue. Metals are not biodegradable and tend to accumulate in soils, posing potential risks to surrounding ecosystems and human health. Plant-based techniques (phytotechnologies) for the in situ remediation of metal-polluted soils have been developed, but these have some limitations. Phytotechnologies are a group of technologies that take advantage of the ability of certain plants to remediate soil, water, and air resources to rehabilitate ecosystem services in managed landscapes. Regarding soil metal pollution, the main objectives are in situ stabilization (phytostabilization) and the removal of contaminants (phytoextraction). Genetic engineering strategies such as gene editing, stacking genes, and transformation, among others, may improve the phytoextraction potential of plants by enhancing their ability to accumulate and tolerate metals and metalloids. This review discusses proven strategies to enhance phytoextraction efficiency and future perspectives on phytotechnologies.

## 1. Introduction

Due to industrial, mining, and agricultural activities, increasing soil HM concentrations have become an urgent global problem [1,2]. Although HMs are natural compounds, anthropogenic activities are major causes of biogeochemical alterations, increasing HM soil concentrations far above natural levels. Land contamination with HMs poses serious risks to human health and ecosystems [1,3]. In humans, high levels of HMs in living tissues cause severe organ impairment, neurological disorders, and eventual death [4]. On the other hand, a high concentration of HMs in soil decreases microbial and plant populations, diversity, and ecosystem functioning [2].

Understanding the soil availability of any metal to plants is complex and multifactorial because one must consider the interactions between HMs and other soil components along with the species-specific capacity to extract metals from soils [5]. Indeed, soil metal toxicity to plants depends on a metal’s soil bioavailability, which varies according to several factors, such as the pH, presence of competitive cations, and content of soil organic matter (SOM), among others [6,7].

Exposure to high HM concentrations can cause severe effects for plant growth and development, such as photosynthesis inhibition, the disruption of cell membrane integrity, root browning, interveinal chlorosis, and, finally, wilting and death [8,9,10]. All of these effects result from the production of reactive oxygen species (ROS) such as superoxide (O_2_^•−^), hydrogen peroxide (H_2_O_2_), and hydroxyl radicals (OH^•^) via Haber–Weiss and Fenton reactions [10,11]. ROS damage essential cellular components such as DNA (degradation), proteins (denaturation), and lipids (oxidation). Thus, the induction of ROS production leads to oxidative stress, affecting plant growth, seed germination, plant biomass production, root length, and chlorophyll biosynthesis. Moreover, various other physiological activities are also adversely affected, such as mineral nutrition, respiration, photosynthetic activity, enzymatic reactions, and alterations of the antioxidant system [10,11,12,13].

In general, plants are sensitive to elevated concentrations of bioavailable fractions of HMs in soils [14]. However, some plants, known as metallophytes, have developed tolerance mechanisms to cope with HM stress [15]. Some metallophytes, known as hyperaccumulators, have also developed mechanisms to accumulate particularly elevated levels of some HMs in their aerial tissues, which may be several hundred or thousand times greater than that in normal plants (two to three orders of magnitude more than what is normally found in plants growing in soils that are not enriched with particular metals) [14]. Metallophytes have antioxidant strategies to cope with oxidative injury induced by HMs. These include ROS-removing enzymes such as superoxide dismutase (SOD), catalases (CAT), guaiacol peroxidase (GPX), ascorbate peroxidase (APX), and glutathione reductase (GR), as well as low molecular mass antioxidant scavengers such as ascorbate (ASC) and glutathione (GSH). Metallophytes have also developed certain HM tolerance mechanisms, including metal exclusion, metal accumulation, metal chelation, and the binding of metals by strong ligands, such as cysteine-rich proteins including metallothioneins (MTs) and thiol-rich peptides, called phytochelatins (PCs) [11,13,16].

## 2. Phytoextraction Technology

The last three decades have seen the emergence and development of environmentally friendly in situ soil remediation techniques using plant species known as phytotechnologies. Phytotechnologies are generally considered less invasive, more cost-effective, friendlier to the environment, and more restorative of soil than conventional methods such as chemical and physical remediation [17,18]. Phytoremediation is a technology based on plants (i.e., trees, shrubs, herbs, and grasses) and their associated microorganisms (i.e., bacteria and arbuscular mycorrhizal fungi (AMF)), used to remove (phytoextraction), degrade (phytodegradation) or immobilize (phytostabilization) toxic substances in environmental matrices such as soil and water [3]. Phytoremediation takes advantage of the ability of certain plants to absorb, accumulate, metabolize, volatilize, or stabilize contaminants in soil, air, water, or sediments [19]. For example, plants can reduce bioavailable concentrations of soil contaminants such as HMs (Pb, Zn, Cd, Cu, Ni, Hg), metalloids (As, Sb), inorganic compounds (NO_3_^−^ NH_4_^+^, PO_4_^3−^), radioactive isotopes, hydrocarbons (i.e., diesel), or pesticides and herbicides (atrazine, bentazone, chlorinated and nitroaromatic compounds), thus restoring soil functions [20,21].

Phytoextraction refers to the capacity of certain plants to uptake (remove) contaminants from the soil by the plant roots system, and the subsequent translocation and accumulation of these into the shoot or any harvestable part of the plant [22,23]. Hyperaccumulator plants can be used to extract metals as well as inorganic and organic pollutants from soil [24,25]. These plants can accumulate HMs in the range of 0.01–1% dry weight in their aerial tissues [21,26]. Specifically, a hyperaccumulator plant can concentrate more than 10 mg kg^−1^ Hg; 100 mg kg^−1^ Cd; 1000 mg kg^−1^ Co, Cr, Cu, or Pb; or 10,000 mg kg^−1^ Zn or Ni [27,28]. The use of hyperaccumulators to clean up metal-polluted soils (bioavailable fraction) has been proposed [29]. However, naturally occurring hyperaccumulators are generally slow-growing plants that produce relatively small amounts of harvestable above-ground biomass [30], thus limiting their phytoextraction potential.

### 2.1. In Situ Phytoextraction Application

In recent years, many remediation technologies, including physical, chemical, biological, and combined methods have been proposed and adopted to mitigate soil contamination [31]. In the case of phytoextraction technology, selecting the appropriate plant species is one of the most important considerations. The appropriate plant species should be capable of tolerating high HM levels and other limiting soil conditions, such as high acidity, salinity, or alkalinity [32]. In some cases, such as in semiarid mining regions, plant species should also be able to adapt to drought and high light radiation. Therefore, metal-tolerant native plants are often selected because they demonstrate tolerance to local environmental conditions and could easily grow and proliferate [33].

In situ phytoextraction with legumes has been regarded as an eco-friendly way for rehabilitating tailings dumps. Studies by Yu et al. (2019) used legumes (*Pongamia pinnata*) to analyze changes to microbial structures during phytoextraction. They monitored dynamic changes to the microbiota in the rhizosphere of *Pongamia pinnata* during a two-year on-site remediation of vanadium–titanium magnetite tailings. After remediation, overall soil health conditions significantly improved: available N and P contents increased, enzyme activities were found and microbial carbon and nitrogen content also increased. This study indicated that legume phytoremediation can effectively cause microbial communities to shift in favor of rhizobia in HM-contaminated soil [34].

Even ornamental flowers or herbs can have phytoextraction potential [35,36]. A field study was conducted to evaluate the efficacy of lavender for the phytoremediation of contaminated soils. The experiment was performed on an agricultural field contaminated by the non-ferrous-metal works near Plovdiv, Bulgaria. Concentrations of Pb, Zn, and Cd in lavender (roots, stems, leaves, and inflorescences) and in the essential oils of lavender were assessed. Lavender is an HM-tolerant plant that can be grown on contaminated soils and can be referred to as a Pb hyperaccumulator and a Cd and Zn accumulator. This ability can be successfully used for the phytoremediation of HM-contaminated soils. It was shown that soil HMs do not influence lavender development or the quality and quantity of lavender essential oil. The possibility of further industrial processing will make lavender an economically interesting crop for farmers of phytoextraction technology [37].

Crop co-planting is widely used in agriculture because it can increase total crop yields through increased resource use efficiency [38]. Xiong et al. (2018) studied the phytoextractive effects of co-planting *Ricinus communis* or legumes (*Medicago sativa*) in Cd- and Zn-contaminated soil. A factory relocation site in Shanghai contaminated with Cd and Zn was selected for the experiment. According to the results of a potential ecological risk assessment of HMs, the study area was divided into three levels of pollution: slight, moderate, and high. The results showed that the presence of *Medicago sativa* can significantly increase the height and biomass of *R. communis*, and there was a greater impact on the chlorophyll content of *R. communis* at higher pollution levels. Differences in pollution levels could significantly change the oil content of *R. communis* plants, but *M. sativa* can alleviate the impact of HMs. The presence of *M. sativa* increased the amount of Cd and Zn in *R. communis* by 1.14 and 2.19 times, respectively. Thus, co-planting *R. communis* and legumes remediated contaminated soil and may be a practical in situ phytoextraction strategy for HM-contaminated soil [39].

### 2.2. Advantages and Limitations of Phytoextraction

The effective phytoextraction of soil metal pollutants depends on three major factors: (1) efficient metal uptake and translocation to the aerial parts of the plant; (2) the ability to accumulate and tolerate high levels of metal; and (3) a well-developed root system and abundant shoot biomass production. Traditional phytoextraction techniques face certain limitations, such as the long time required for soil remediation (e.g., several crop seasons), remediation being restricted to soil layers where roots can develop, and limited extraction ability due to small above-ground biomass production [21]. Another limitation is that only a small metal soil fraction is normally bioavailable to plants (bioavailable fraction) [40]. Finally, the technique is only applicable to sites with low or moderate metal pollution [21,41].

Recently, approaches based on chemically or microbiologically assisted phytoextraction techniques have been suggested to improve soil metal remediation, particularly at large scales [42]. To cope with some of the plant limitations for proper soil metal phytoextraction, genetic engineering tools may be used to develop transgenic plants with higher aerial biomass production and increased metal tolerance and accumulation capabilities that are also well adapted to a variety of climatic conditions [43]. With genetic engineering strategies such as gene editing, stacking genes, transformation, and the overexpression of strategic genes, among others, it may be possible to improve a plant’s potential to accumulate and tolerate HMs. This review will discuss some strategies for improving phytoextraction efficiency.

## 3. Genetic Engineering Strategies to Enhance Phytoextraction Efficiency

### 3.1. Enhancement of Metal Accumulation

Metal homeostasis is essential for plant growth, development, and adaptation to diverse environmental stressors [44]. Therefore, plants have specific transporters to tightly regulate the uptake, distribution, and utilization of metal ions and thus maintain redox homeostasis [45,46,47]. New biotechnological techniques have made it possible to better understand plant molecular mechanisms and enhance them through genetic engineering. For example, the gene editing of plant individuals allows the improvement of certain capacities or abilities [48]. Genome editing can make phytoextraction technologies more efficient, time-saving, and economically feasible, minimizing limitations and ensuring large-scale application [42]. The genes that are currently widely used to improve plant phytoextraction potential are those that encode transporters of metal ions [49]. The overexpression of key proteins such as metal-binding proteins or metal transporters could enhance the uptake and accumulation of HMs. Among transporters, various families play important roles in maintaining redox homeostasis, including members of the Zn/Fe-regulated transporter (ZRT/IRT-related ZRT-IRT-like proteins (ZIP)) family, natural resistance-associated macrophage protein (NRAMP) family, cation diffusion facilitator (CDF) family, yellow stripe-like (YSL) family, major facilitator super (MFS) family, P1B-type heavy metal ATPase (HMA) family, vacuolar iron transporter (VIT) family, and the cation exchange (CAX) family, among others [50,51,52].

#### Overexpression of Metal Transporters

Shim et al. (2013) produced genetically engineered *Bonghwa poplar* (*Populus alba x P. tremula* var*. glandulosa*) lines expressing the yeast ScYCF1 gene (*Saccharomyces cerevisiae*-yeast cadmium factor 1), which encodes a vacuolar transporter involved in toxic metal sequestration into the vacuole. When grown on HM-polluted soil from a mining site, ScYCF1-expressing plants showed reduced Cd toxicity symptoms and accumulated more Cd in comparison to wild plants (*WT*) [53]. When plants were tested in contaminated soil, root dry weight and the accumulation of Cd, Zn, and Pb in transgenic roots were higher than in *WT*, demonstrating a potential utilization for these lines in long-term phytoextraction and the phytostabilization of highly contaminated soils [15,53].

ZIP genes represent an important family of transporters that are highly conserved among species (plans, fungi, and animals) [54]. Members of this gene family are responsible for transporting a variety of cations, including Fe, Mn, and Zn [54]. They may also be involved in the transportation of non-essential and highly toxic HMs, such as Cd. Jiang et al. (2021) who explored the function of an *SmZIP* gene isolated from *Salix matsudana* and its role in Cd tolerance, uptake, translocation, and distribution [55]. By overexpressing the *SmZIP* transporter in transgenic tobacco, they found that Cd-stress-induced phytotoxic effects were reduced compared to *WT* plants. Moreover, compared to *WT* tobacco, the Cd content of roots, stems, and leaves in the transgenic tobacco increased, and the Zn, Fe, Cu, and Mn contents also increased. Furthermore, the transgenic *SmZIP* tobacco exhibited a higher growth rate and showed a more vigorous phenotype. The overexpression of *SmZIP* resulted in the redistribution of Cd at the subcellular level, a decrease in the percentage of Cd in the cell walls, and an increase in Cd in the soluble fraction of both roots and leaves. Thus, the overexpression of *SmZIP* plays important roles in Cd accumulation and translocation, subcellular distribution, and chemical forms in transgenic tobacco under Cd stress [55].

Another well-studied family of transporters is the ATP-binding cassette transporter family, such as *AtATM3*. *AtATM3* is localized at the mitochondrial membrane of *Arabidopsis thaliana* and is involved in the biogenesis of Fe–S clusters and Fe homeostasis in plants. Through *Agrobacterium*-mediated genetic transformation, Bhuiyan et al. (2011) overexpressed the *AtATM3* gene into *Brassica juncea* (Indian mustard), a plant species suitable for phytoextraction. *AtATM3* overexpression in *B. juncea* conferred enhanced tolerance to Cd (II) and Pb (II) stressors. Transgenic seedlings showed a significant increase in the accumulation of both Cd (II) and Pb (II). The enhanced HM tolerance and accumulation by *AtATM3* transgenic plants was attributed to higher *BjGSHII* (*B. juncea* glutathione synthetase II) and *BjPCS1* (phytochelatin synthase 1) expression levels induced by *AtATM3* overexpression. Hence, *AtATM3* transgenic plants are more tolerant to HMs and can accumulate more HMs to enhance phytoextraction in contaminated soils [56]. Similarly, L. Sun et al. (2018) isolated an ATP-binding cassette (ABC) transporter gene *PtABCC1* from *Populus trichocarpa* and overexpressed it in *Arabidopsis* and poplar. Transgenic plants possessed higher Hg tolerance than *WT* plants, and the overexpression of *PtABCC1* led to a 26–72% increase in Hg accumulation in *Arabidopsis* and a 7–31% increase in poplar leaves and 26–160% increase in the poplar stem. These results demonstrated that *PtABCC1* plays a crucial role in the tolerance and accumulation of Hg in plants and is thus a suitable strategy for improving Hg phytoextraction [57].

Another example of the increase in HM uptake capacity in plants is the enhancement of Cd phytoextraction by the overexpression of a cation diffusion facilitator (CDF family), or a metal tolerance/transport protein (MTP family). Das et al. (2016) isolated and functionally characterized the *OsMTP1* gene from Indica rice (*Oryza sativa* L. cv. IR64) to study the potential application of this transporter to improve the efficiency of Cd phytoextraction [46]. The heterologous expression of *OsMTP1* in tobacco resulted in a reduction in Cd stress-induced phytotoxic effects, including growth inhibition, lipid peroxidation, and cell death. Compared to the *WT*, transgenic tobacco plants showed enhanced vacuolar thiol content, indicating the vacuolar localization of sequestered Cd. The transgenic tobacco plants exhibited significantly higher biomass production (2.2–2.8-fold) and hyperaccumulation (1.96–2.22-fold) of Cd compared to *WT* under Cd exposure. Transgenic plants also showed moderate tolerance and accumulation to As under As exposure. These results suggest that transgenic tobacco plants overexpressing *OsMTP1* could be useful in future phytoextraction applications for cleaning up Cd-contaminated soils, as is also shown by the results of Bhuiyan et al. (2011) [56] and Sun et al. (2018) [57].

In plants, Cu acts as an essential cofactor of numerous proteins that perform central functions in cells. Because Cu is an essential micronutrient, plants have specific mechanisms not only to exclude or chelate it but also to transport it into cells [58]. The predominant Cu transportation mechanism is the reduction in the ion by plasma membrane NADPH-dependent cupric reductases [59] and the subsequent uptake of the metal by high-affinity Cu+ transporters of the COPT family under the control of the Cu-responsive transcription factor SPL7 (SQUAMOSA promoter-binding protein-like7) [59,60,61,62]. Studies performed by Andrés-Colás et al. (2010) showed that transgenic plants overexpressing either COPT1 or COPT3 exhibited increased intracellular Cu levels and were sensitive to Cu in the growth medium [63]. Similarly, Sanz et al. (2019) expressed the COPT2 transporters in a heterologous system, such as oocytes of the African clawed frog *Xenopus laevis*. They observed that the Cu content in oocytes expressing the COPT2 transporter increased, accumulating over 6-fold more Cu than control oocytes. These results suggest that the overexpression of COPT high-affinity transporter family proteins can increase plant Cu uptake capacity [61].

The uptake and translocation of non-essential HMs in plants occur through metal transporters of essential micronutrients such as the natural resistance-associated macrophage protein (NRAMP). NRAMPs from different species exhibit different biological functions, although their sequences share similarities [64,65,66]. Wang et al. (2019) isolated an NRAMP6 from *Ailanmai* (*Triticum turgidum* L. ssp. *turgidum*) that encoded a plasma membrane protein. Expressing TtNRAMP6 in yeast significantly enhanced Cd concentration, and therefore the cells were more sensitive to Cd. Furthermore, the overexpression of TtNRAMP6 increased Cd concentration in roots, stems, leaves, and the whole plant of Arabidopsis, which indicated that overexpression of TtNRAMP6 enhanced Cd accumulation [62].

### 3.2. Strategies to Enhance Metal Tolerance

Plants have developed a number of mechanisms to detoxify excess metals, such as compartmentalization in inactive tissues, chelation by metal ligands, and detoxification by antioxidants. Metal chelators such as organic acids, amino acids, phytochelatins, and metallothioneins play important roles in metal detoxification [67].

#### 3.2.1. Overexpression of Metal-Binding Proteins

Metallothioneins (MTs) are low-molecular mass, cysteine-rich proteins that are broadly distributed in microorganisms, plants, and animals. These proteins can bind metals and form complex biochemical structures [68]. MTs play a fundamental role in metal homeostasis, detoxification, and reactive oxygen species (ROS) scavenging [69]. Numerous studies have observed the expression of plant MTs in response to various HM stressors, including Cu, Zn, and Cd stress. The overexpression of some MTs has led to enhanced metal tolerance [69]. The overexpression of *Elsholtzia haichowensis* metallothionein 1 (EhMT1) in tobacco plants enhances Cu tolerance and accumulation in root cytoplasm, decreasing hydrogen peroxide (H_2_O_2_) production [68]. Another strategy for improving HM tolerance is the overexpression of the ThMT3 gene (*Tamarix hispida* metallothionein 3 in *Salix matsudana*), which has been found to increase Cu tolerance, nitric oxide (NO) production, and NO release, which contributes to the induction of adventitious roots under Cu stress. The application of NO has been shown to induce the transcription and accumulation of MTs in leaves, indicating the possible functions of NO and MTs in response to HMs. NO is an important gas-signaling molecule involved in many developmental and physiological processes, including defense responses against toxic metals in plants [70].

Gu et al. (2014) isolated a full-length cDNA homolog of *MT2a* (type 2 metallothionein) from the Cd-tolerant species *Iris lactea* var. *chinensis*. The expression of *IlMT2a in I. lactea* var. *chinensis* roots and leaves was upregulated in response to Cd stress [71]. When the gene was constitutively expressed in *A. thaliana*, the roots of transgenic lines were longer than those of *WT* under 50 μM or 100 μM Cd stress. However, there was no difference in Cd absorption between *WT* and transgenic lines. Transgenic lines accumulated remarkably less H_2_O_2_ and O_2_^•−^ (superoxide ion) than *WT*. These results indicate that *IlMT2a* may be a promising gene for the improvement of Cd tolerance in plants. Similarly, Zhang et al. (2014) isolated a type 2 metallothionein gene, *SaMT2*, from the Cd/Zn co-hyperaccumulator *Sedum alfredii* Hance [72]. The ecotype was a Zn/Cd hyperaccumulator discovered in an old Pb/Zn mining area in China [73] which can accumulate up to 9000 mg g^−1^ Cd and 29,000 mg g^−1^ Zn in shoots without symptoms of toxicity [74]. This large amount of metals in plant cells requires a powerful detoxification system to protect plants from deleterious effects. *SaMT2* encodes a putative peptide of 79 amino acid residues, including two cysteine-rich domains. The transcript level of *SaMT2* was higher in the shoots than in the roots of *S. alfredii* and was significantly induced by Cd and Zn treatments. Expressed in yeast, *SaMT2* significantly enhanced Cd tolerance and accumulation. Ectopic expression of *SaMT2* in tobacco enhanced Cd and Zn tolerance and accumulation in both the shoots and roots of transgenic plants. By expressing the metallothionein gene *SaMT2*, transgenic plants showed higher antioxidant enzyme activities and accumulated less H_2_O_2_ than *WT* plants under Cd and Zn treatment. Hence, *SaMT2* could significantly enhance Cd and Zn tolerance and accumulation in transgenic tobacco plants by chelating metals and improving the antioxidant system.

The cell number regulator 2 (TaCNR2) from common wheat (*Triticum aestivum*) is similar to plant Cd resistance proteins involved in regulating HM translocation. To understand the effect of TaCNR2 on HM tolerance and translocation, K. Qiao et al. (2019) overexpressed the TaCNR2 gene in *Arabidopsis* and rice. A real-time quantitative PCR indicated that TaCNR2 expression in wheat seedlings increased under Cd, Zn, and Mn treatment. The overexpression of TaCNR2 in *Arabidopsis* and rice enhanced their tolerance to Cd, Zn, and Mn, and overexpression in rice improved Cd, Zn, and Mn translocation from roots to shoots. Results showed that TaCNR2 can transport HM ions. Thus, this study provides a novel gene resource for increasing nutrient uptake and reducing toxic metal accumulation in crops [75].

Another strategy for improving phytoextraction efficiency is the overexpression of phytochelatins (PCs). Phytochelatins play important roles in the detoxification and tolerance of HMs in plants [76]. The synthesis of PCs is catalyzed by phytochelatin synthase (PCS), which is activated by HMs [77]. Zhu et al. (2021) isolated a PCS gene, *BnPCS1*, from the bast fiber (defined as fibers obtained from the outer cell layers of the stems of various plants) of the crop ramie (*Boehmeria nivea*) using the RACE (rapid amplification of cDNA ends) method. The *BnPCS1* promoter region contains several cis-acting elements involved in phytohormone or abiotic stress responses. Subcellular localization analysis indicates the fact that the *BnPCS1*-GFP protein localizes in the nucleus and the cytoplasm. Real-time PCR assays showed that the expression of *BnPCS1* was significantly induced by Cd and the plant hormone abscisic acid (ABA). Transgenic lines that overexpressed the *BnPCS1* gene exhibited better root growth and fresh weight, lower levels of MDA and H_2_O_2_, and higher Cd accumulation and translocation compared to the *WT* under Cd stress. Taken together, these results could provide new gene resources for the phytoextraction of Cd-contaminated soils [78].

#### 3.2.2. Overexpression of Enzymes

Kumar et al. (2019) generated transgenic alfalfa (*Medicago sativa* L.) plants that overexpressed the *Arabidopsis* ATP sulfurylase gene using *Agrobacterium*-mediated genetic transformation. Selected transgenic lines showed increased tolerance to a mixture of five HMs (Pb, Cd, Cu, Ni, and V) as well as demonstrated enhanced metal uptake abilities under controlled conditions. The transgenic lines were fertile and did not exhibit any apparent morphological abnormalities. The results of this study indicated an effective approach for improving the HM accumulation ability of alfalfa plants, which can then be used for the remediation of metal-contaminated soils in arid regions [79].

Ascorbate peroxidase (APX) plays an important role in oxidative stress metabolism in higher plants [80]. Xu et al. (2008) analyzed the role of APX in protecting against excessive-Zn-induced oxidative stress in transgenic *Arabidopsis* plants constitutively overexpressing a peroxisomal ascorbate peroxidase gene (*HvAPX1*) from barley. They found that transgenic plants were more tolerant to Zn stress than *WT* plants. Under Zn stress, the concentrations of hydrogen peroxide and malondialdehyde (MDA) accumulation were higher in *WT* plants than in transgenic plants. Therefore, the mechanism of Zn tolerance in transgenic plants may be due to reduced oxidative stress damage. Under Zn stress, activities of APX were significantly higher in transgenic plants than in *WT* plants. The authors also found that Zn accumulation in shoots was much higher in transgenic plants than in *WT* under Zn stress. In addition, transgenic plants were more tolerant to excessive Cd stress and accumulated more Cd in shoots than *WT*. These results suggest that *HvAPX1* plays an important role in Zn and Cd tolerance and might be a candidate gene for developing high-biomass-tolerant plants for the phytoextraction of Zn and Cd in metal-polluted environments [81].

## 4. New Strategies for Phytoextraction

### 4.1. Bio-Assisted Phytoextraction

Microbial-assisted phytoremediation is a promising strategy for hyperaccumulating, detoxifying, or remediating soil contaminants. Arbuscular mycorrhizal fungi (AMF) are found in association with almost all plants, contributing to healthy performance and providing resistance against environmental stressors. AMF colonize plant roots and extend their hyphae to the rhizosphere region, assisting in the mineral nutrient uptake and regulation of HM acquisition as well as growth enhancement by nutrient acquisition, detoxification of MH, secondary metabolite regulation, and enhancement of abiotic/biotic stress tolerance [82].

Sun et al. (2017) studied the different conditions of bioremediation of Pb-contaminated soil using *Solanum nigrum* L. combined with *Mucor circinelloides*. They observed that, when compared with a control, Pb removal efficacy was optimal with a microbial/phytoremediation strategy, compared with phytoremediation only, which in turn was a better approach than microbial remediation. The bioremediation rates were 58.6, 47.2, and 40.2% in microbial/phytoremediation, microbial remediation, and phytoremediation groups, respectively. Inoculating soil with *M. circinelloides* enhanced Pb removal *and S. nigrum* L. growth. Furthermore, soil fertility increased after bioremediation according to changes in enzymatic activities. The results indicated that inoculating *S. nigrum* L. with *M. circinelloides* enhanced its efficiency for the phytoremediation of soil contaminated with Pb [83].

Another example of arbuscular mycorrhizal fungi-assisted phytoextraction is a study conducted by Singh et al. (2019). In this study, arbuscular mycorrhizal fungi (AMF) were used to promote the growth of *Zea mays* L. in HM-rich tannery sludge (HMRTS). To identify suitable AMF species, a pot experiment was conducted using *Rhizophagus fasciculatus*, *Rhizophagus intraradices*, *Funneliformis mosseae*, and *Glomus aggregatum* for the cultivation of *Zea mays* L. under HMRTS. AMF treatments significantly influenced plant growth and phytoremediation potential. Interestingly, *F. mosseae* acted as a bio-filter in roots and modulated the direct translocation of HMs (Cd, Cr, Ni, Pb) and micronutrients from soil to shoot (bioaccumulation factor) as well as roots to shoots (Translocation factor) in plants. In HMRTS, AMF inoculation was also found to significantly improve soil microbial enzymatic activities, such as dehydrogenase, β-Glucosidase, and acid and alkaline phosphatase. The finding of this study suggests that AMF-assisted cultivation of *Zea mays* is a promising approach for the phytoremediation of HMRTS [84].

The application of beneficial soil microbes is gaining significant attention. El-Esawi et al. (2020) investigated the role of *Serratia marcescens* BM1 in enhancing the Cd stress tolerance and phytoextraction potential of soybean plants. The inoculation of Cd-stressed soybean plants with *Serratia marcescens* BM1 significantly enhanced plant growth, biomass, gas exchange, nutrient uptake, antioxidant capacity, chlorophyll content, total phenolics, flavonoids, soluble sugars, and proteins. Moreover, *Serratia marcescens* BM1 inoculation reduced the levels of Cd and oxidative stress markers but significantly induced the activities of antioxidant enzymes and the levels of osmolytes and stress-related gene expression in Cd-stressed plants. Furthermore, the application of 300 μM CdCl_2_ and *Serratia marcescens* triggered the highest expression levels of stress-related genes. Overall, this study suggested that the inoculation of soybean plants with *Serratia marcescens* BM1 promotes phytoextraction potential and Cd stress tolerance by modulating photosynthetic attributes, osmolytes biosynthesis, antioxidants machinery, and the expression of stress-related genes [85].

Microbial-assisted phytoextraction was used to enhance hyperaccumulation, detoxification, and the remediation of soil contaminants. The use of either arbuscular mycorrhizal fungi (AMF) or bacteria has been proven to be beneficial for plants, assisting in mineral nutrient uptake, regulation of HM acquisition, growth and root systems, and abiotic/biotic stress tolerance. Overall, microbial-assisted phytoextraction is a suitable strategy for enhancing phytoremediation technology, but it has some limitations in terms of the level of pollution that can be successfully applied.

### 4.2. Epigenetic Regulation

In recent years, several studies have elucidated the different signal transduction pathways involved in HM responses, identifying complementary genetic mechanisms conferring tolerance to plants [86]. The regulation of HM-responsive genes has been related to epigenetic mechanisms such as DNA methylation and histone modifications, which can repress or activate gene expression through DNA modification as well as by avoiding transposon movement. It has been demonstrated that the DNA hypermethylation of the genome is involved in the HM stress response by protecting DNA from possible damages caused by metal subproducts [87].

Among cytotoxic ions, the trivalent aluminum cation (Al^+3^) formed by the solubilization of aluminum (Al) into acid soils is one of the most abundant and toxic elements under acidic conditions. Specific genes related to Al tolerance, measured in contrasting tolerant and susceptible rice varieties, exhibited differences in DNA methylation frequency. The differential methylation patterns could be associated with the epigenetic regulation of rice responses to Al stress, highlighting the major role of epigenetics over specific abiotic stress responses [86,88].

Feng et al. (2016) studied the variation of DNA methylation patterns associated with gene expression in rice (*Oryza sativa*) exposed to cadmium. They reported genome-wide single-base resolution maps of methylated cytosine and transcriptome change in Cd-exposed rice [89]. Widespread differences were identified in CG and non-CG methylation marks between Cd-exposed and Cd-free rice genomes. More hypermethylated than hypomethylated genes were found, and many of the genes were involved in stress response, metal transport, and transcription factors. Most DNA methylation-modified genes were transcriptionally altered under Cd stress. A study by Niedziela (2018) showed similar results. Liquid chromatography (RP-HPLC), methylation amplified fragment length polymorphisms (metAFLP), and methylation-sensitive amplification polymorphisms (MSAP) analysis were used to investigate the effects of aluminum (Al) stress on DNA methylation levels in the crop species triticale. RP-HPLC, but not metAFLP or MSAP, revealed significant differences in methylation between Al-tolerant (T) and non-tolerant (NT) triticale lines. The direction of methylation change was dependent on the plant phenotype and organ. Al treatment increased the level of global DNA methylation in roots of T lines by approximately 0.6%, whereas demethylation of approximately 1.0% was observed in NT lines. DNA methylation in leaves was not affected by Al stress. The metAFLP and MSAP approaches identified DNA alterations induced by Al^3+^ treatment [90].

Another example of epigenetic regulation is the ubiquitination process. Ubiquitin (Ub)-extension protein (UBQ) functions as a Ub-donor in the Ub/26S proteasome system, which is widely engaged in degrading target proteins and thus participates in a broad range of physiological responses [91,92]. Ubiquitination-dependent protein degradation is involved in plant growth, development, and environmental interactions, but the functions of ubiquitin-ligase (E3) genes are largely unknown in tomatoes (*Solanum lycopersicum* L.). Similarly, Ahammed et al. (2020) functionally characterized a RING E3 ligase gene, *SlRING1*, that positively regulates Cd tolerance in tomato plants. An in vitro ubiquitination experiment showed that *SlRING1* has E3 ubiquitin ligase activity. The determination of subcellular localization revealed that *SlRING1* is located in the plasma membrane and the nucleus. The overexpression of *SlRING1* in tomatoes increased the chlorophyll content, the net photosynthetic rate, and the maximal photochemical efficiency of photosystem II (Fv/Fm), but reduced the levels of reactive oxygen species and relative electrolyte leakage under Cd stress. Moreover, *SlRING1* overexpression increased the transcript levels of catalase (CAT), dehydroascorbate reductase (DHAR), monodehydroascorbate reductase (MDHAR), glutathione (GSH1), and phytochelatin synthase (PCS), which contributed to the antioxidant and detoxification response. Crucially, *SlRING1* overexpression also reduced concentrations of Cd in both shoots and roots. Thus, enhanced tolerance to Cd due to induced SlRING1-overexpression is attributed to reduced Cd accumulation and the alleviation of oxidative stress. These findings suggest that *SlRING1* is a positive regulator of Cd tolerance, which could be a potential breeding target for improving HM tolerance in plants [93].

### 4.3. Gene Stacking

Guo et al. (2008) studied the development of transgenic plants with increased HM tolerance and accumulation by simultaneous overexpression of AsPCS1 and GSH1 (derived from garlic and baker’s yeast) in *A. thaliana*. Phytochelatins (PCs) and glutathione (GSH) are the main binding peptides involved in chelating HM ions in plants and other living organisms [67,76,94]. Single-gene transgenic lines had a higher tolerance to Cd and As and accumulated more Cd and As than the *WT*. Compared to single-gene transgenic lines, dual-gene transformants exhibited significantly higher tolerance to Cd and As and accumulated more Cd and As. One of the dual-gene transgenic lines, PG1, accumulated twice as much Cd as single-gene transgenic lines. The simultaneous overexpression of *AsPCS1* and GSH1 led to elevated total PC production in transgenic *Arabidopsis*. The results indicate that stacking modified genes increases Cd and As tolerance and accumulation in transgenic lines and represents a highly promising new tool to be used in plant-based remediation efforts.

Similar results were obtained when Zhao et al. (2014) isolated a gene-encoding PC synthase (*PaPCS*) and tested its function through heterologous expression in a strain of yeast sensitive to Cd. Subsequently, a Cd-sensitive and high-biomass-accumulating species, *Festuca arundinacea*, was transformed, either with *PaPCS* or *PaGCS* (a glutamyl cysteine synthetase gene of *Phragmites australis*) individually (single transformants) or with both genes together in the same transgene cassette (double transformant). The single and double transformants showed greater Cd tolerance and accumulated more Cd and PC than *WT* plants, and their Cd leaf/root ratio content was higher. Thus, *PaGCS* appears to exert a greater influence than *PaPCS* over PC synthesis and Cd tolerance and accumulation. The double transformant has interesting potential for phytoextraction [77].

Another example is *Mulberry* (*Morus* L.), one of the most ecologically and economically important tree genera, which has the potential to remediate HM-contaminated soils. Fan et al. (2018) identified two *Morus notabilis PCS* genes based on a genome-wide analysis of the *Morus* genome database. Quantitative real-time polymerase chain reaction (qRT-PCR) analysis revealed that, under 200 μM Zn^2+^ stress or either 30 or 100 μM Cd^2+^ stress, a relative expression of each of the two *MaPCSs* (from *Morus alba*) was induced in root, stem, and leaf tissues within 24 h of exposure to metals, with Cd2+ inducing expression more strongly than the Zn^2+^ overexpression of *MnPCS1* and *MnPCS2* in *Arabidopsis* and tobacco enhanced Zn^2+^/Cd^2+^ tolerance in most transgenic individuals. The results of transgenic *Arabidopsis* lines overexpressing *MnPCS1* and *MnPCS2* suggest that *MnPCS1* plays a more important role in Cd detoxification than *MnPCS2*. In addition, there was a positive correlation between Zn accumulation and the expression level of *MnPCS1* or *MnPCS2*. Results indicated that *Morus PCS1* and *PCS2* genes play important roles in HM stress tolerance and accumulation, providing a useful genetic resource for enhancing tolerance to HMs and increasing the HM phytoextraction potential of these plants [95].

LeDuc (2004) studied the overexpression of selenocysteine methyltransferase (SMT) in *Arabidopsis* and Indian mustard to increase selenium (Se) tolerance and accumulation. SMT detoxifies selenocysteine by methylating it to methylselenocysteine, a nonprotein amino acid, thereby diminishing the toxic misincorporation of Se into protein. The authors used genetic engineering to develop fast-growing plants with an increased ability to tolerate, accumulate, and volatilize Se by incorporating a gene encoding the *selenocysteine methyltransferase* from the Se hyperaccumulator *Astragalus bisulcatus* into Indian mustard. The resulting transgenic plants successfully enhanced Se phytoextraction by tolerating and accumulating Se significantly better than *WT* plants. In order to enhance the phytoextraction of selenate, LeDuc (2004) developed double transgenic plants that overexpressed the gene-encoding ATP sulfurylase (APS) in addition to SMT. Results showed that there was a substantial improvement in Se accumulation from selenate (a 4–9 times increase) in transgenic plants overexpressing both APS and SMT [96].

### 4.4. Gene Editing and Genetic Engineering

In recent years, an innovative gene editing technique called the CRISPR–Cas9 system has been developed. This technique is commonly used for gene knockout experiments and to edit the genomes of a diverse range of crop plants [97]. It consists of a Cas9 nuclease, which creates a DNA double-strand break (DSB), and a guide RNA (gRNA), which is responsible for directing the nuclease to a specific region in the genome. The endogenous repair mechanisms of cells can lead to gene deletion. Genetic engineering could use important genes identified through transcriptomics to develop ideal hyperaccumulator plants. Incorporating advanced technologies such as CRISPR–Cas9 and synthetic genes will help enhance phytoextraction technology [98]. Moreover, Tang et al. (2017) demonstrated the ability of CRISPR to reduce Cd accumulation in rice by knocking out the metal transporter gene, *OsNramp5*. This is perhaps the most significant advance of CRISPR in phytoextraction to date and highlights the promise of its use in gene transcription regulation [99].

Nevertheless, an area of CRISPR research that may hold even greater potential for phytoextraction improvement is the use of gRNA-guided dCas9 to modulate gene expression. Transcription factors can be fused with dCas9 to repress or enhance transcription by RNA polymerase and subsequently upregulate or downregulate the expression of a gene or a group of genes of interest [100]. The CRISPR–Cas9 system has been adapted to generate technology called CRISPRa (CRISPR Activation). CRISPRa uses an inactivated Cas9 nuclease (dCas9) that cannot generate DNA double strand disruption to target genomic regions, resulting in RNA-directed transcriptional control. Cas9 can be fused with different transcriptional activation domains that can be targeted to promoter regions by the guide RNA (gRNA), which recruits additional transcriptional activation domains to upregulate the expression of the target gene [101]. By using the CRISPRa system, a catalytically dead dCas9 fused to a transcriptional activator peptide can increase transcription of a specific gene, through a designed gRNA sequence to direct the dCas9-activator to promoter or regulatory regions of the gene of interest [102,103].

Editing genes using recent techniques such as CRISPR–Cas can enhance the natural capacity of a plant to grow, accumulate, and tolerate HMs, though this is not considered a transgenic approach. CRISPR–Cas9 seems to be a more promising technique for modifying gene expression without introducing foreign genes. Taking all of this into consideration, the modification of gene expression, metabolic pathways, and pollutant homeostasis networks that support hyperaccumulation, tolerance, or degradation could be used to enhance the HM uptake efficiency of plants while avoiding metal toxicity. Therefore, gene editing and genetic engineering are considered a suitable strategy for enhancing the phytoextraction process.

### 4.5. Use of Native Plants as a Study Model

The standard approach for dealing with the limitations of phytoextraction technology is to use genes characterized from tolerant or hyperaccumulator exotic plants in model (traditional) species with fast growth rates and a significant production of aerial biomass. However, the use of invasive, non-native species can affect biodiversity [104]. Some of these plant species can intrude into the surrounding natural areas, thereby causing the disruption and alterations of ecosystem functions, reducing native biodiversity, and negatively impacting local economies and human well-being [105].

A more suitable solution for enhancing in situ phytoextraction efficiency could be to use native plants that are already acclimatized to the abiotic stress caused by HMs in the soil. Choosing the appropriate plant species is a critical step in correctly implementing any in situ phytotechnologies (e.g., phytoextraction, phytostabilization, and phytomining). Therefore, using and modifying native or endemic plants that grow in contaminated sites could be a better strategy for enhancing phytoremediation efficiency.

In addition, native plants that naturally colonize metal-polluted sites are an important source of metal-tolerant microorganisms that can be used in bio-assisted phytoremediation. The aquatic fern *Azolla filiculoides* Lam. (*Salviniaceae*) is an efficient metal hyperaccumulator that possesses an endophytic microbiome with PGPB potential [106].

Depending on site-specific conditions (i.e., climate), metal-enriched soils could coexist with additional co-occurring stressors, such as drought and salinity, which can further restrict phytoextraction [107]. In these cases, a combination of two or more abiotic stressors may occur and result in a new condition for plant development, different from the effect of each stressor by itself [108,109]. Thus, plant selection for phytoextraction must also consider the presence of multiple co-occurring stressors and their effects on plant growth and development [110]. Abandoned mine tailings sites are a global problem, with thousands of unvegetated, exposed tailings piles presenting a source of contamination for nearby communities. Tailing disposal sites in arid and semiarid environments are especially subject to wind dispersion and water erosion [111,112,113]. Establishing plant species on mine tailings in arid and semiarid regions is impeded by physicochemical factors including extreme temperatures, low precipitation, high winds, low nutritional contents, and high salt concentrations [113], among others, thus constituting a number of co-occurring plant stressors. Lam et al. (2017) selected three native plant species, *Prosopis tamarugo*, *Schinus molle*, and *Atriplex nummularia*, to be used in a study of the phytoremediation potential of native plants growing on a copper mine tailing in northern Chile. The plants were selected because of their natural presence in northern Chile and their capacity to grow in sites with similar characteristics to those of the mine tailing under consideration [114]. Additional examples of this study of combined stressors on plants performance include the studies of Orrego (2020) which evaluated the effect of single and combined Cu, NaCl, and water stress on the growth parameters of three Atriplex species with phytostabilization potential: *Atriplex atacamensis*, *A. halimus*, and *A. nummularia*. *Atriplex* species are typical of dry and salty soils. This study showed that the *Atriplex* species are differentially affected by salt, drought, and metal stress and that combined stress causes an overall negative effect on growth parameters [115].

There are many studies that search and identify metal-tolerant plants, i.e., metallophyte and hyperaccumulator ecotypes, growing in contaminated sites such as industrial and agricultural soils with elevated metal concentrations [116]. In Latin America, metallic ores are abundant and diverse. Because of wealth mineral deposits, polluted areas, weather conditions, and unique plant diversity, metal-tolerant and hyperaccumulator plants (metallophytes) are likely to be found in this region [117]. However, because scientific research on metallophytes has been scarce in Brazil, Cuba, Dominican Republic, Venezuela, Argentina, Paraguay, and Chile, few metal-tolerant and metal-hyperaccumulator plants have been reported in Latin America in comparison with other areas of the world [117,118,119]. To date, 172 plant species have been described as either metal tolerant (30 species) or hyperaccumulators (142 species), a low number when compared to the high diversity of plant species in the region [120]. Recently, mercury (Hg) accumulation capacity was assessed in three plant species (*Axonopus compressus*, *Erato polymnioides*, and *Miconia zamorensis)* that grow on soils polluted by artisanal small-scale gold mines in the Ecuadorian rainforest. Researchers found consortia interaction between arbuscular mycorrhizal fungi (AMF) and these plant species. For example, *E. polymnioides* increased Hg accumulation when grown with greater AMF colonization [121].

Sugarcane-molasses distillery waste (sludge) contains not only a mixture of complex organic pollutants but also a high quantity of Fe (5264.49), Zn (43.47), Cu (847.46), Mn (238.47), Ni (15.60), and Pb (31.22 mg kg^−1^) which enhances its toxicity to the environment. Chandra and Kumar (2017) evaluated the phytoextraction pattern of 15 native plants growing in post-methanated distillery sludge (PMDS). The investigators studied the phytoextraction potential of native weeds and grasses. This study showed that from the selected plants, *Blumea lacera*, *Parthenium hysterophorous*, *Setaria viridis*, *Chenopodium album*, *Cannabis sativa*, *Basella alba*, *Tricosanthes dioica*, *Amaranthus spinosus* L., *Achyranthes* sp., *Dhatura stramonium*, *Sacchrum munja*, and *Croton bonplandianum* were root accumulators for Fe, Zn, and Mn. *S. munja*, *P. hysterophorous*, *C. sativa*, *C. album*, *T. dioica*, *D. stramonium*, *B. lacera*, *B. alba*, *Kalanchoe pinnata*, and *Achyranthes* sp. were found to be shoot accumulators for Fe. In addition, *A. spinosus* L. was found to be a shoot accumulator for Zn and Mn. These results indicated the high accumulation and translocation capabilities of these plants. Furthermore, ultrastructural observations of root tissues revealed deposits of HMs in various cellular components without any apparent toxic effects [122,123]. Hence, these native plants may be used as a tool for in situ phytoremediation and the eco-restoration of industrial-waste-contaminated sites.

Another example of a native hyperaccumulator species is the perennial herb *Phytolacca acinosa* Roxb. (*Phytolaccaceae*), which is found in southern China. Field surveys on Mn-rich soils and glasshouse experiments have found that this herb is a manganese hyperaccumulator. This species not only has remarkable tolerance to Mn but also has extraordinary uptake and accumulation capacity for this element. These results confirm that *P. acinosa* is an Mn hyperaccumulator that grows rapidly and has substantial biomass, wide distribution, and broad ecological amplitude. This species provides a new plant resource for exploring the mechanism of Mn hyperaccumulation and has potential for use in the phytoremediation of Mn-contaminated soils [124].

Another study, led by Amer (2013), showed three endemic Mediterranean plant species, *Atriplex halimus*, *Portulaca oleracea*, and *Medicago lupulina*, which were hydroponically grown to assess their potential use in phytoremediation and biomass production. *Atriplex halimus* and *M. lupulina* produced high shoot biomass with relatively low metal translocation to the above-ground parts of the plants. Plant metal uptake efficiency ranked as follows: *A. halimus* more efficient than *M. lupulina*, and the latter being more efficient than P. oleracea. Due to the high biomass production and relatively high metal content in the roots, *A. halimus* and *M. lupulina* could be successfully used in phytoremediation and specifically in phytostabilization [125].

Ke et al. (2007) studied two *Rumex japonicus* populations, one from a Cu mine and the other from an uncontaminated site. The researchers conducted growth experiments under hydroponic conditions to evaluate Cu accumulation and mineral nutrient content under excess Cu and nutrient deficiency conditions [126]. The tolerance indices of the contaminated population were significantly higher than the uncontaminated population, indicating the evolution of Cu tolerance in the contaminated population. At control levels and low levels of Cu treatment, there was no difference in Cu accumulation in the roots of the two populations. At high Cu (100 μM) treatment, however, the contaminated population accumulated less Cu in roots than the uncontaminated one, suggesting root exclusion mechanisms in acclimatized plants. Plants with exclusion strategies are currently used to revegetate bare soil areas with high metal concentrations; plants with an accumulation strategy are used for the phytoextraction of high-metal soils [127]. *Rumex japonicus* plants from a Cu mine heap use the exclusion strategy and could potentially be used to recover vegetation in Cu-contaminated soil areas. Compared with those in uncontaminated sites, the plants of *R. japonicus* growing at a Cu-contaminated mine site presented higher growth rates, Cu tolerance, and mineral nutrient deficiency tolerance. Furthermore, their mineral composition was less affected by Cu stress, suggesting that the stability and homeostasis of mineral composition under nutrient deficiency stress plays an important role in the Cu tolerance of plants [126].

*Polypogon australis* Brong. (Poaceae) is a native grass of Chile that spontaneously colonizes abandoned Cu mine tailings deposits and accumulates Cu in leaves and roots at levels considered phytotoxic for other plant species [128]. Ortiz-Calderón (2008) found that *Polypogon australis* growing on mine tailings had 670 and 223 mg kg^−1^ Cu (dry matter) in leaves and roots, respectively. The total content of Cu in plant tissues was 892.5 mg kg^−1^ (dry weight), and the leaves-to-root ratio of the Cu content was 3.0, suggesting Cu translocation from roots to leaves [129]. Jara-Hermosilla et al. (2017) characterized the status of H_2_O_2_-reducing enzyme activity in the facultative metallophyte species *Polypogon australis* when treated with a mining liquid waste (MLW) derived from a copper mine. To determine the effect of the solubility of metals present in the MLW, the researchers studied the accumulation of elements, variations in H_2_O_2_ and lipoperoxidation levels, and the relationship of theses parameters with the H_2_O_2_ reduction activity in *P. australis* plants at pH 5.1 (acidic MLW) and pH 6.7 (neutral MLW) for two weeks. The results showed that the metal content of the MLW—but not the solubility of the metals—provoked an increase in the H_2_O_2_ content in the plants tissues and triggered the enzymatic control of H_2_O_2_ [130]. Noni-Morales et al. (2019) evaluated the ability of *P. australis* to germinate and grow in soil contaminated with diesel oil. *Polypogon australis* plants germinating and growing in diesel-polluted soils exhibited high tolerance and survival compared with other diesel-tolerant species. The calculated effective concentration (EC_50_) of diesel for *P. australis* was 4.5%. *Polypogon australis* germinated and grew on all diesel concentrations used in the experiments. The species was classified as tolerant to diesel oil [131].

Metal-tolerant plants and other stress-tolerant plants are well represented in the Poaceae family [132]. Shalmani et al. (2019) studied the B-box (BBX) proteins that play important roles in plant growth regulation and development, impacting photomorphogenesis, the photoperiodic regulation of flowering, and responses to biotic and abiotic stress [133]. These researchers retrieved a total of 131 BBX members from five Poaceae species, including 36 from maize, 30 from rice, 24 from sorghum, 22 from stiff brome, and 19 from millet. They observed changes in the expression patterns of BBX members in response to abiotic, hormonal, and HM stress, showing the B-box protein potential roles in plant growth and development and in responding to multivariate stresses. Findings suggested that BBX genes could be used as potential genetic markers for plants, particularly in functional analysis and under multivariate stressors. Ezaki et al. (2013) produced a model for Al tolerance in *Andropogon virginicus* (Poaceae). Collectively, their results suggested that *A. virginicus* showed high Al tolerance, with a combination of five independent approaches: (1) the suppression of Al uptake by the roots from the soil; (2) high Al transportation from root to shoot; (3) accumulation and secretion of Al in leaves; (4) induction of anti-peroxidation enzymes and polyphenols by Al; and (5) Al-induced NO production in roots [134].

The utilization of native plants has advantages and disadvantages. In terms of advantages, native plants are already adapted to environmental stressors due to natural selection and evolution and are tolerant to the multiple stressors of the site. Moreover, some metallophytes are herbaceous plants that therefore have a fast growth rate. A key disadvantage of using native plants, however, is that they have unknown genomes, and protocols for transformation and in vitro regeneration must be defined for them.

## 5. Legal and Normative Limitations

Genome editing consists of producing directed, permanent, and inheritable mutations at a specific place in the genome, mediated by DNA repair systems in the cell, with the lowest probability of committing unwanted errors (off-targets) and leaving no foreign DNA sequences. New plant breeding technologies (NPBTs) such as Zn finger nucleases (ZFN), transcriptional activator-like effector nucleases (TALEN), clustered regularly interspaced short palindromic repeats associated with the Cas9 endonuclease (CRISPR–Cas9), oligo-directed mutagenesis (ODM), cisgenesis, RNA-directed DNA methylation (RdDM), grafting, reverse breeding, and agroinfiltration have been used to induce specific mutations in the genome, to introduce beneficial traits, or to express transgenes in a specific tissue in a wide range of crops and model plants [100,135].

Gene-editing technology, such as CRISPR–Cas9, holds great promise for the progression of science and applied technologies. This foundational technology enables the modification of the genetic structure of any living organism with unprecedented precision [136]. The recent development and scope of the CRISPR–Cas system have raised new regulatory challenges worldwide due to moral, ethical, safety, and technical concerns associated with its applications in pre-clinical and clinical research, biomedicine, and agriculture [137]. However, in order to enhance its potential for societal benefit, it is necessary to adopt rules and adequate regulations. This requires an interdisciplinary effort in legal thinking. Any legislative initiative needs to consider both the benefits and the ethical aspects of gene editing from a broad societal and value-based perspective [136].

Different countries have different regulations for the approval and cultivation of crops developed using NPBTs such as gene editing [138]. Plant breeding technologies have expanded, accelerating breeding research beyond the confines of current regulations. The application of genome editing, such as CRISPR–Cas9, does not neatly fit into existing regulatory frameworks, creating uncertainty as to whether they can be used as conventionally developed varieties without further regulation [139]. In general, the analysis focuses on whether a “new combination” of genetic material has occurred. This is defined as “a stable insertion of one or more genes or DNA sequences that encode proteins, interfering RNA, double-stranded RNA, signaling peptides or regulatory sequences” [140].

In Chile, the procedures for the import, domestic propagation, and re-export of propagated genetically modified (GM) plant material in the country were established through Extent Resolution 1523/2001. Agricultural and Livestock Service (SAG) from Chile establishes standards for the internment and introduction to the environment of live modified plant organisms of propagation under the Resolution 1523/2001. This is the key criterion determining whether an organism will be considered as a genetically modified organism (GMO) and whether Resolution 1523/2001 should apply to new materials derived from NPBTs. In this resolution, a GMO is defined as “a living biological organism, capable of transferring or replicating genetic material, including the sterile organism, viruses, and viroids that possesses a new combination of genetic material that has been obtained through the application of modern biotechnology.” This regulatory framework allows GM seed production exclusively for the export and research and development activities; the permanence and commercialization of GM seeds are not allowed [140].

The European Union (EU) defines GMOs as an organism, with the exception of human beings, in which the genetic material has been altered in a way that does not naturally occur by mating and/or natural re-combination. EU legislation on environmental issues that aims to protect the environment are established by a mix of regulations, which directly apply in member states, and directives, which set the framework in the relevant area but are then transposed by member states into national law. The Directive 2001/18/EC on the deliberate release of GMOs into the environment and the Directive 2015/412 to restrict or prohibit the cultivation of genetically modified organisms in their territory, are an example of the use of framework regulations in the EU. This directive aims to protect human health and the environment when: 1. releasing GMOs into the environment for any purpose, such as experimental use; and 2. placing GMOs on the market as a or part of a product. Deliberate release into the environment means any intentional introduction into the environment of a GMO or a combination of GMOs for which no specific containment measures are used to limit their contact with and to provide a high level of safety for the general population and the environment [141].

Even if a country has a stable protocol of regulation and legislation of NPBTs, every case must be individually analyzed. Governments should consider the regulatory framework of genome editing technologies and establish appropriate regulations, if necessary, without creating obstacles to the commercialization of products derived from these technologies. Nevertheless, most countries have no legislation whatsoever and must create a legislation frame for the use of NPBTs. Regulatory frameworks need to be further developed to effectively channel the power of technology and direct it to beneficial applications. Humanity should be kept in the driving seat rather than being trampled by technology [136].

## 6. Conclusions

In this review, we discuss the advantages and limitations of different strategies for enhancing HM accumulation and tolerance by the genome editing and transformation of metallophyte plants (i.e., hyperaccumulator and metallophyte native plants) with phytoextraction potential.

The best candidate genes for improving the process of phytoextraction that we discussed in this review are metallothionein (MT), phytochelatin (PC), phytochelatin synthase (PCS), metal transporters, and antioxidant-related genes. These genes enhance plant performance in soil polluted with HMs. They increase metal uptake capacity and accumulation, antioxidant activity, and translocation and compartmentalization, leading to increased metal tolerance and accumulation. Thus, the combined overexpression of metal transporter and metal-binding genes with antioxidant-related genes is a significant strategy for developing high-biomass-tolerant plants for phytoextraction. Overall, the strategies mentioned above to enhance accumulation indirectly enhance tolerance, and vice versa, indicating that these processes are tightly interconnected.

Other interesting strategies include the use of epigenetic regulation, gene stacking, and gene editing. Although a single-gene transgenic strategy is very effective for increasing metal tolerance and accumulation, the phytoextraction capacity of plants is multifactorial. Overall, a combination of approaches that modifies more than one gene (i.e., stacking genes) is more effective than a single transformation, showing great potential to enhance phytoremediation efficiency. Additionally, the implementation of transgenic plants has legal and normative limitations.

Accordingly, selecting the appropriate plant species is one of the most important considerations in the in situ phytoextraction process. Native plants that already grow on contaminated sites have the highest potential to simultaneously be great candidates for phytoextraction and revegetation. Overall, plants that naturally grow and colonize sites with high metal concentrations are the best candidates for study, not only as a source of target genes but also as study models.

Using NPBTs such as CRISPR to improve the phytoextraction capacity of native plants seems to be the most promising strategy for phytoextraction technologies to reach their greatest potential and reduce the environmental risk. CRISPR-aided genome engineering shows potential for exploiting plant genomes to enhance phytoremediation. The CRISPR–Cas system is the most versatile genome-editing tool in the history of molecular biology because it can be used to alter diverse genomes (e.g., genomes from both plants and animals), including human genomes, with unprecedented ease, accuracy, and efficiency. Future research must be focused on the use of NBPTs to enhance the plant growth and biomass production, transport, metabolization, and compartmentalization of HMs, and root system development, among others, to increase their phytoremediation potential. Furthermore, since CRISPR is such a versatile tool, we can target multiple genes or traits at the same time, achieving higher efficiency genome editing and saving time and resources, leaving no molecular trace. The regulatory and normative frameworks for NPBTs must not become obstacles to developing genome editing technologies that are beneficial for the environment and public health. It is possible to improve plant-based technologies for cleaning HM polluted environments, and concurrently, to recover elements of economic interest.

## Data Availability

Not applicable.

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
