# Peer review of "Increase in Phytoextraction Potential by Genome Editing and Transformation: A Review"

_plants, 2021, doi:10.3390/plants11010086_

Round 1

Reviewer 1 Report

  1. L25-45: The description of this paragraph is irrelevant to the subject of this article, and it is recommended to delete it.
  2. The section "5. In situ phytoextraction" does not seem to be related to the subject of this article, nor is it connected to the previous and subsequent section. It is recommended to delete it.
  3. The content of the conclusion is too long. Many narratives are repeated with the previous content. It is recommended to simplify, and put forward suggestions and possible aspect for future research. 
  4. In the section "6. Legal and normative limitations", the author has provided a detailed explanation. It is recommended that the author add a section to compare the cost-effectiveness of traditional methods with genome editing and transformation methods, including the effect of improvement and the time required for improvement, etc. 
  5. Many research literature citations do not follow the Type of the paper.

Reviewer 2 Report

The text is well readable and understandable, though I am basically an analytical chemist. There is a lot of information, and I have some questions about the structure of the information given.

What is the main difference between chapter 2.1 and 4.5? Both treat the use of native plants as a study model for metal tolerance

Chapter 3 is about how to favour the formation of metal binding molecules. Differences between 3.1.1 and 3.1.2 are not clear; I would prefer to sort according to the target organ: vacuole - mitochondria - cell wall - inert precipitate. 3.1.2 and 3.2.1 have the same headline - please change!

How about excluders? They enrich the site with organic carbon, which lowers metal availability and dilutes the metalliferous substrate.

Chapter 4 is about how to change the genes from special plants (?)

Chapter 5 is about usual crop plants (?)

Please state differences between the chapters more distinctly!

Chapter 6 needs some additions. Usually, the reader is not so interested in the legislation of Belize, than what is relevant for the European Union or the US. Most restrictions are for human and animal nutrition, but read carefully:

Council Directive 90/220/EEC on the deliberate release into the ernvironment of genetically modified organisms

Directive 2015/412 to restrict or prohibit the cultivation of genetically modified organisms in their territory
